# Graphene Materials from Coke-like Wastes as Proactive Support of Nickel–Iron Electro-Catalysts for Water Splitting

**DOI:** 10.3390/molecules29061391

**Published:** 2024-03-20

**Authors:** María González-Ingelmo, Victoria G. Rocha, Zoraida González, Uriel Sierra, Enrique Diaz Barriga, Patricia Álvarez

**Affiliations:** 1Instituto de Ciencia y Tecnología del Carbono (INCAR), Consejo Superior de Investigaciones Científicas CSIC, Francisco Pintado, Fe 26, 33011 Oviedo, Spain; maria.ingelmo@incar.csic.es (M.G.-I.); vgarciarocha@incar.csic.es (V.G.R.); zoraidag@incar.csic.es (Z.G.); 2Laboratorio Nacional de Materiales Grafénicos, Centro de Investigación en Química Aplicada, Blvd. Enrique Reyna Hermosillo, 140, Saltillo 25294, Mexico; uriel.sierra@ciqa.edu.mx (U.S.); enriqe.diazbarriga@ciqa.edu.mx (E.D.B.)

**Keywords:** waste, graphene, 3D electrode, electrocatalysis, NiFe, water splitting, OER

## Abstract

Graphene materials, used as electrocatalyst support in green hydrogen production, contribute to increasing the efficiency and robustness of various systems. However, the preparation of a hybrid catalyst containing graphene materials from industrial wastes is still a challenge due to the heterogeneity of the waste. We report the synthesis of 3D electrodes using graphene oxides (GOs) from industrial waste (IW) prepared by immersion onto Toray carbon paper as a 3D support onto GO suspensions and electrodepositing NiFe layered double hydroxides (LDHs). Standard graphite was also used as the reference. The morphology of the two hybrid electrodes was determined by SEM, HRTEM, XPS. Although very similar in both, the sample containing graphene from IW (higher Csp^3^ hybridization in the graphene layer) has a NiFe phase with less crystallinity and larger presence of Fe^2+^ ions. These electrodes exhibited similar activity and stability as electrocatalysts of the oxygen evolution reaction (OER), demonstrating the proactive effect of the graphene into the 3D electrode even when this is prepared from heterogeneous industrial waste. Moreover, the defective graphenic structure of the waste GO enhances the reaction kinetics and improves the electron transfer rate, possibly due to the small differences in the electrodeposited NiFe LDH structure.

## 1. Introduction

Global population growth and economic expansion are causing a sharp rise in the energy demand, which in turn is causing serious climate change. The need for renewable energy systems is rising as a result of the continuous restructuring of traditional economies and industrial sectors to solve these problems [1,2]. This has led scientists to hunt for more affordable and ecologically friendly energy sources [3,4]. One of the green alternatives as a clean energy source is hydrogen, which constitutes an alternative to conventional fossil fuels [5,6]. Water splitting has been shown to be an efficient method for creating hydrogen, a clean energy source [7,8,9]. The process involves two fundamental half-reactions, which are a hydrogen evolution reaction (HER) and oxygen evolution reaction (OER). Due to their slow kinetics and high overpotential, especially in the case of OER, it is necessary to employ electrocatalysts. Nonprecious metals like Ni, Fe, or Co, and also their alloys (NiFe, etc.), are frequently employed as electrocatalysts for the OER in an alkaline electrolyte [10]. One of the most promising options among them is NiFe layered double hydroxide (LDH), which is widely thought to be the best option due to its plentiful catalytic sites, layered and open structure, being obtained also at an affordable price [11,12]. However, the electron transport step is severely restricted by its poor conductivity and small surface area, which leads to a decline in electrochemical performance and limits the step of the electron transport. This leads to the degradation of electrochemical performance, which is the main limitation of the use of NiFe LDH catalysts [13,14]. Furthermore, obtaining affordable electrodes without sacrificing their activity, durability, or long-term performance is one of the main hurdles in OER research [15,16,17,18,19]. The proactive effect of carbon nanomaterials (i.e., graphenes) in OER catalytic activity is well known. They can act as catalysts themselves by controlling surface defects [20,21] or can be used as proactive supports, contributing to improving catalyst dispersion, acting as co-catalysts or modulating the catalysts’ properties [22]. However, the large-scale production of graphene technologies is being limited by their progress, mostly because of their expensive cost. Typically, graphenes are prepared from graphites [23,24,25,26,27]. They can also be prepared from other fossil sources, such as pre-graphitic materials [28,29]. The composition and the structure showed by these precursors are, in all cases, homogeneous (due to the homogeneity in the natural resources—in the case of natural graphite—or due to the controlled synthetic procedure in which they are obtained in the case of synthetic graphite or coke). This facilitates the control of the composition and structure of the obtained graphenes, particularly when they are produced by top-down technologies (controlling the oxidation/reduction conditions or by appropriately choosing the crystallinity of the starting graphite [30,31,32]). The result is the tuning of their final surface characteristics which, in turn, could favor their active role as supports in a variety of catalytic systems [33,34].

A more environmentally friendly approach to graphene production consists of using, as a carbon source, a carbonaceous residue, like plastic waste [35], or other carbonaceous sources [36,37]. In line with these ideas, we have recently reported the preparation of graphene oxides from industrial waste which consists of a residue of the production of coke in a coke oven battery (which is deposited at the top inner section of the coking oven) [38]. This industry belongs to an energy technology dependent on coal or coke, immersed nowadays in a deep transformation to be carried out in the future. However, while moving towards a more sustainable technology, it is essential to minimize the actual environmental problems, for example managing and/or reusing their waste [39]. The main issue in this preparation, as in the utilization of other wastes, is their heterogeneity. In fact, the obtained graphene oxide from this industrial waste revealed a complex and highly defective graphenic structure, as a result of the highly heterogeneous composition of the raw industrial waste. This presence of defects was demonstrated to modify the reactivity of the waste-based graphene towards the aryl diazonium salt chemistry [40] in the preparation of hybrid NCH-Ir/graphene materials [41]. However, we also demonstrated that, when using an adequate procedure, these defects can be used to improve the OER catalytic activity of the hybrid system.

Given these promising results, we study herein the utilization of graphene oxide from industrial waste in the preparation of 3D electrodes doped with nickel–iron LDH and their utilization in OER electrocatalytic reactions. The electrodes were prepared by a sequential method that is initiated with the immersion of fibrous paper (Toray carbon paper) into suspensions of graphene oxide and subjecting them to the electrodeposition of the nickel/iron LDH, which acts as a catalyst. The chemical and morphological features of the 3D structures are evaluated and compared to that of a similar 3D material prepared from graphite-based GO. This allows for the correlation of each catalytic OER performance with the intrinsic properties of each material.

## 2. Results

### 2.1. Preparation of Hybrid Graphene/NiFe 3D Electrodes from GO from a Coke-like Wastes

Industrial waste (IW), containing an average of ~11wt.% of ashes, was previously transformed into graphene oxide (GO-IW) by a modified Hummers method that removes the initial ashes [38]. Relative to graphene oxide prepared from graphite (GO-G), it contains a larger number of sp^3^ hybridized defective carbon atoms (7.8% vs. 5.6% in GO-G, Table 1) and a larger Raman ID/IG ratio (1.07 vs. 0.95 in GO-G, Table 1, and Appendix A in the Appendix A). This is consistent with a more defective graphenic structure.

The 3D electrodes were prepared using as a support fibrous material (Toray carbon paper, TCP) and the GOs previously described. The first step is to immerse the TCP, previously treated at 550 °C to improve its wettability, in an aqueous suspension (2000 ppm) of the corresponding graphene oxide materials (GO-G and GO-IW). The as-obtained carbon papers with graphene oxide were subsequently thermally reduced at 400 °C to enhance adherence to the fibers and their conductivity. SEM images confirm that the processing leads to GO-IW films homogeneously distributed among the TCP fibers (Figure 1a,b). Similar results were obtained for the GO-G, which shows the graphene-like film coating carbon fibers in TCP at a similar size and thickness (Figure 1c,d) [38].

This appearance is maintained after electrodepositing the nickel–iron species for both electrodes (TCP-GO-IW-NiFe, Figure 1e,f and TCP-GO-G-NiFe, Figure 1g,h). It should be noted that no nanoparticles were observed on the carbon network through SEM microscopy. In any case, the presence of these metals is confirmed using SEM-EDX (see Appendix A). The intensity of the signals corresponding to Ni and Fe in EDX was, however, low. This could be attributed to the formation of an extremely thin layer on the surface, which makes it barely detectable by the SEM-EDX beam (penetrating tens of nanometers into the material and resulting in the detection of the carbon electrode on which this metallic nanosheet is presented). This is consistent with previous results [42] demonstrating the creation of an extremely thin nanosheet comprising hydroxides of Ni and Fe. According to the literature, the reduction in the nitrate precursor results in the formation of hydroxides, which would then cause an increase in pH. As a result, bimetallic hydroxides would form on the surface of the electrode due to the reaction of Ni^2+^ and Fe^3+^ cations. It is also worth mentioning that this effect is similar for the two GOs studied, with no influence from the morphology of the GO used.

HRTEM analysis of both electrodes allows for the visualization of the proposed NiFe structures formed. The TEM-EDX spectra (Appendix A) confirm the presence of Ni and Fe in both samples, with similar results as observed in the SEM-EDX analysis. Additionally, in both cases, the electron-dense regions corresponding to the NiFe appear homogeneously distributed along the surface of the graphene layer (Figure 2), but some differences can be noticed. The electron diffraction patterns (SAED, insets in Figure 2a,b) of each of those selected show the distinct lattice plane spacing (0.15, 0.17, and 0.25 nm) corresponding to the (100), (200), and (012) planes of the NiFe [43]. However, in the case of TCP-GO-IW-NiFe, it is only possible to observe two of these LDH NiFe planes, (100) and (200), which could indicate the formation of a less crystalline NiFe LDH phase. Moreover, the crystalline phase corresponding to the graphene lattice is more visible in the case of TCP-GO-G-NiFe (Figure 2a, inset), possibly suggesting that in TCP-GO-IW-NiFe, the graphene layer is more covered by the laminar structure of the NiFe LDH. This could be related to the larger number of Csp^3^ carbon atoms in the materials from industrial waste [41]. In any case, the STEM-EDX images and mapping of the electrodes show a homogeneous distribution of the Ni, Fe, and O atoms all along the graphene basal planes (see Appendix A).

In order to gain further information on the Ni and Fe nanoparticles deposited, the quantification of each metal into the 3D electrodes was calculated by ICP determinations (Table 2). As observed, the relative percentage of Ni and Fe in each 3D electrode varies depending on the GO used as a raw material, making a Ni/Fe ratio of 6.8 for TCP-GO-IW-NiFe and 5.3 for TCP-GO-G-NiFe. This indicates that variations in the surface chemistry of the carbon materials deposited on the TCP lead to different Ni and Fe electrodeposition during the preparation of the 3D electrode. It also implies that the presence of defective graphenic carbon atoms (Csp^3^, which are more abundant in the industrial waste 3D electrode, as shown in Table 1) promotes the electrodeposition of a greater number of Ni atoms in the resulting NiFe LDH.

For the in-depth study of the chemical structure of the 3D electrodes, XPS general spectra (Appendix A) and high-resolution XPS characterization were measured and analyzed (Figure 3). No differences were observed in the Ni2p spectra of the material containing graphene from reference graphite or industrial waste (Figure 3a). In both cases, the typical features for predominantly Ni^2+^ species in the LDH were shown, with maxima values at 854 and 871 eV for the first doublet [44]. However, slight differences could be seen for the Fe 2p spectra. The maxima feature in the Fe 2p spectrum of TCP-GO-G-NiFe is found at 710.6 eV. This is typically attributed to Fe^3+^ species in the LDH. However, TCP-GO-IW-NiFe also exhibits a high intensity at relatively lower binding energies (709.3 eV). This could be related to the presence of additional Fe^2+^ species in this sample, in comparison to TCP-GO-G-NiFe. These species of Fe^2+^ could be interfering in the NiFe LDH by substituting Ni^2+^ atoms, which could cause the formation of Fe-O-Fe motifs [45]. This modification could potentially produce changes in the crystalline structure, as was observed in the TEM analysis.

### 2.2. Electrocatalytic OER Activity of the as-Prepared 3D TCP/Graphene/NiFe Electrodes

The as-prepared 3D TCP doped with graphene and NiFe electrodeposited electrodes was tested in the electrocatalytic OER reaction as a working electrode in a Teflon home-made three-electrode cell (see Appendix A for details). For comparison, electrodes prepared with the electrodeposited NiFe in the absence of GO (TCP-NiFe) and only TCP were also tested (Figure 4).

In the linear sweep voltammetry (LSV) experiments shown in Figure 4a, it can be observed that coating the fiber with all the GOs results in an improvement in the overpotential value at 10 mAcm^−2^ in comparison to TCP with Ni and Fe electrodeposited in the same way but without prior GO impregnation (Figure 4a, brown line). These differences in catalysis when impregnating the TCP with carbon material could be related to an increase in the overall surface area (corresponding to the graphene surface), as seen in the SEM images (Figure 1). This could promote the greater electrodeposition of Ni and Fe and also confirm the proactive effect of graphene in these 3D systems. It is also interesting to mention that the 3D systems without electrodeposited Ni and Fe do not show substantial catalytic activity, indicating the poor activity of graphene materials themselves (see Appendix A). By evaluating the LSV curves obtained for the two 3D electrodes containing graphite (Figure 4a, blue and red lines), it can be observed that both exhibit a very similar profile, with similar overpotential and activity, which is higher than that of the 3D electrode without graphene (Figure 4a, brown line). Regarding LSV curves, it can also be observed that the Ni^2+/^Ni^3+^ oxidation peak has higher intensity in the case of TCP-GO-IW-NiFe. This could be related to the additional presence of Fe^2+^ species in this sample, which is demonstrated to regulate the local structure and make high-valence metal sites more stable [45]. Additionally, these results in activity towards OER were further confirmed by the Tafel analysis (Figure 4b). The Tafel slope of TCP-GO-IW-NiFe (Figure 4b, red line) is slightly lower than that of TCP-GO-G-NiFe (Figure 4b, blue line) (109.7 mV dec^−1^ vs. 116.7 mV dec^−1^, respectively), which could suggest a positive effect of the defective graphene structure of the industrial waste in the OER reaction mechanism. In any case, both values are significantly lower than the 3D electrode without graphene (132.5 mV dec^−1^ Figure 4b, brown line). The long-term stability of both 3D electrodes containing graphene materials was also tested by CP experiments at 10 mAcm^−2^ for 1h (Figure 4c) and for 24 h (Appendix A), which shows no signal detachment of the NiFe and with a current density remaining constant after 1 h of experiments. Although both samples exhibit stable profiles in CP experiments, there are differences in the required potential to achieve 10 mAcm^−2^ between the two samples, a fact that was not detected in the LSV experiments. These slight variations in favor of TCP-GO-IW-NiFe could be related to differences in surface phenomena affecting reactivity, such as charge transfer resistance. In this sense, the electron transfer kinetics of the graphene-based 3D electrodes were evaluated through electrochemical impedance spectroscopy (EIS) [46]. The Nyquist plots of both electrodes (Figure 4d) show the typical semicircles at the high-frequency range, which are correlated to the charge transfer resistance (R_ct_) of the active materials. They were fitted to the equivalent circuit shown in the inset of Figure 4d, obtaining a R_ct_ value for TCP-GO-IW-NiFe of 1.951 Ω as a result, lower than that obtained for TCP-GO-G-NiFe (5.146 Ω). All this is consistent with an improved electron transfer rate in the electrode containing a more defective carbonaceous graphenic structure (higher number of Csp^3^ atoms, Table 1). In addition to the proactive effect of graphene obtained from waste, this lower R_ct_ value could also be partially caused by the presence of Fe^2+^ species in the TCP-GO-IW-NiFe, as was shown by the XPS characterization. The presence of Fe^2+^ in the NiFe LDH structure is previously reported to positively affect the charge transfer resistance, demonstrating an acceleration of OER kinetics [45].

It is also interesting to highlight the importance that the Fe content has in the catalytic activity of Ni-Fe hydroxides. Several works have previously described the effects of Fe on the composition and structure of the catalyst. In this sense, in ref. [47], it is reported that maximum OER activity is reached with 12–17% Fe content incorporated into the Ni phase, which is the case of both 3D electrodes prepared herein TCP-GO-IW-NiFe and TCP-GO-G-NiFe (12.8 and 15.7%, respectively, Table 2) and explains the good OER catalytic results obtained in both cases. In order to explain the differences between electrodes, we take into consideration the previously discussed electronic similarities between the NiFe LDH obtained for both 3D electrodes. It seems, therefore, that the utilization of a graphene material containing a more defective carbonaceous structure contributes to enhancing the proactive catalytic effect of the overall electrode.

## 3. Experimental Section

### 3.1. Raw Materials and Graphene Oxide Preparation

Industrial coke-like waste was supplied by Industrial Minera México S.A (Nueva Rosita, México, named IW). For comparative purposes, a commercial graphite was used as a graphene precursor. IW contains 11.3% of ashes with composition Al (40–45 wt.%), Fe (39–42 wt.%), Ca (13–15 wt.%) and, at a lower extent, Mg (1–2 wt.%) and Zn (0.5–1.5 wt.%), as determined by ICP-Ms. Graphene oxides were prepared from IW and G by means of a modified version of Hummer’s method [29,48], which is described elsewhere [38], and summarized in the Appendix A.

### 3.2. Preparation of Electrodes Containing Graphene/Ni-Fe Hybrid Catalysts

Graphene materials containing Ni/Fe LDH were prepared using the graphene oxides GO-IW and GO-G previously prepared and subjected to a two-step method. Initially, the procedure comprises a deposition of each of the GOs onto Toray carbon paper (TCP), which acts as the 3D support for the electrode. This TCP (TGP-H-60) was thermally treated at 550 °C for 1 h in air atmosphere. The next step was the immersion of the carbon papers in an aqueous suspension of the corresponding GO (2000 ppm) and subsequent carbonization at 400 °C. The electrodeposition of Ni/Fe LDH was achieved using a three-electrode cell which used the as-prepared TCP/graphene materials as working electrodes. They were immersed in an aqueous suspension of electrolyte containing an equal molar (3 mM) of nickel (II) and iron (III) nitrates [42,49] and subjected to cyclic voltammetry at −1 V (versus RE) for 5 min. This yields the 3D electrodes TCP-GO-r-NiFe (r: G, IW, depending on the GO used).

### 3.3. Electrochemical Measurements

The analysis of the electrochemical performance of the electrodes was evaluated under N_2_ in a custom-made Teflon three-electrode cell (see Appendix A) with KOH 1M as the electrolyte. The experimental details are summarized in the Appendix A.

### 3.4. Scientific Equipment—Characterization of Supports and Hybrid Catalysts

Precursors and samples were characterized by different techniques: X-ray diffraction (XRD, Bruker D8Advance ECO, Bruker, Madrid, Spain) by selecting the radiation frequency of the kα1 line from Cu (1.5418 Å). The power supply was 40 kV and 25 mA. Raman spectroscopy (Xplora- Horiba Scientific, HORIBA Instruments Incorporated, Austin, TX, USA, with argon ion laser of 532 nm). FTIR was conducted on a Thermo Scientific Nicolet iS5 (ICSA. Valencia, Spain, ATR: germanium crystal; integration time of 90 s). X-ray photoemission spectroscopy (XPS, SPECS system, SPECS Group, SPECS Surface Nano Analysis GmbH, Berlin, Germany, pressure of 10–7 Pa. X-ray source of Mg Kα) was evaluated using Gaussian and Lorentzian functions and a Shirley baseline [50] calibrating the C1s line at 284.5 eV. Inductively Coupled Plasma Mass Spectrometry (ICP-MS, Agilent 7700× instrument, Agilent Technologies Spain, Madrid, Spian) [51] was used to evaluate the amount of Ni and Fe in the 3D electrodes. Transmission electron microscopy (HRTEM, FEI-TITAN, FEI Technology de México S.A., Monterrey, Mexico including a 200–300 kV field emission gun microscope, with a symmetrical condenser objective lens type S-TWIN (with a spherical aberration (Cs = 1.25 mm))) was also used.

## 4. Conclusions

This paper demonstrates the capacity of the carbonaceous waste obtained from scrapping the top inner part of coke ovens in the steel industry to be used as a precursor of graphene materials and describes how these can be satisfactorily processed to produce 3D graphene/NiFe electrocatalytic electrodes.

The graphene oxides from industrial waste, compared to other GOs prepared from standard graphite, exhibit a larger number of carbon atoms in sp^3^ hybridization (contributing to the defective graphenic structure). This is due to the heterogeneous nature of the industrial waste and conditions of the subsequent processing of the 3D electrode.

Hybrid graphene/NiFe 3D electrodes were produced by a multi-step procedure comprising the immersion of the TCP in GO aqueous suspensions, partial thermal reduction, and the subsequent electrodeposition of NiFe LDH catalyst.

The similar morphology of the electrode was determined by SEM, HRTEM/SAED, and XPS analysis. Although similar for both samples containing different GOs (i.e., similar Ni/Fe ratio, as determined by ICP/Ms), the use of a graphene obtained from industrial waste (higher number Csp^3^ atoms) contributes to the electrodeposition of a less crystalline LDH NiFe phase, and in which a certain part of the Fe^3+^ ions were substituted by Fe^2+^ ions.

The 3D electrodes were successfully tested as electrocatalysts for the oxygen evolution reaction (OER), an important half-reaction in the water splitting process to obtain green hydrogen, showing high electrocatalytic activity and good long-term stability. In both cases, the proactive effect of the graphene in its composition was confirmed. Moreover, the utilization of graphene from industrial waste contributed to not only improving the reaction kinetics but also to improving the electron transfer rate, possibly due to the small differences in the structure of the electrodeposited NiFe LDH.

The overall results highlight the benefits of processing industrial pre-graphitic waste instead as an alternative to graphite for the preparation of advanced materials for green hydrogen production.

## Figures and Tables

**Figure 1 molecules-29-01391-f001:**
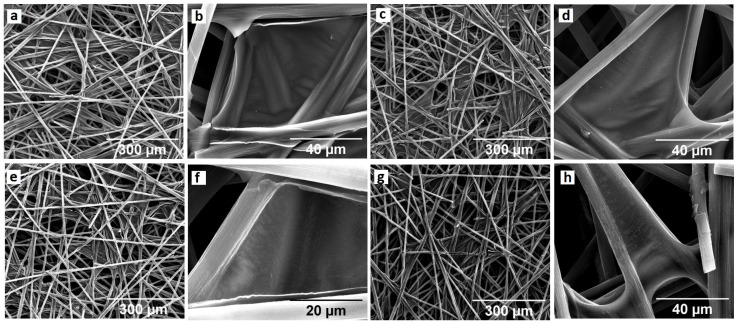
SEM images of TCP-GO-IW (**a**,**b**), TCP-GO-G (**c**,**d**), TCP-GO-IW-NiFe (**e**,**f**), and TCP-GO-G-NiFe (**g**,**h**).

**Figure 2 molecules-29-01391-f002:**
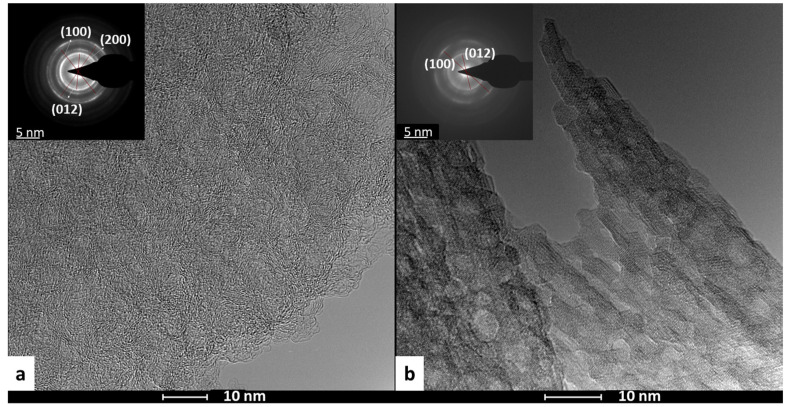
HRTEM of 3D electrodes TCP-GO-G-NiFe (**a**) and TCP-GO-IW-NiFe (**b**), and the insets correspond to SAEDs.

**Figure 3 molecules-29-01391-f003:**
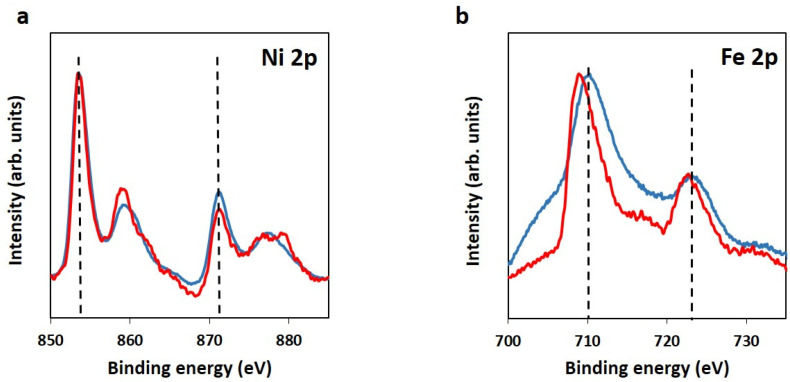
High-resolution Ni 2p (**a**) and Fe 2p (**b**) XPS spectra of TCP-GO-G-NiFe (blue) and TCP-GO-IW-NiFe (red). Selected maxima values as dashed lines.

**Figure 4 molecules-29-01391-f004:**
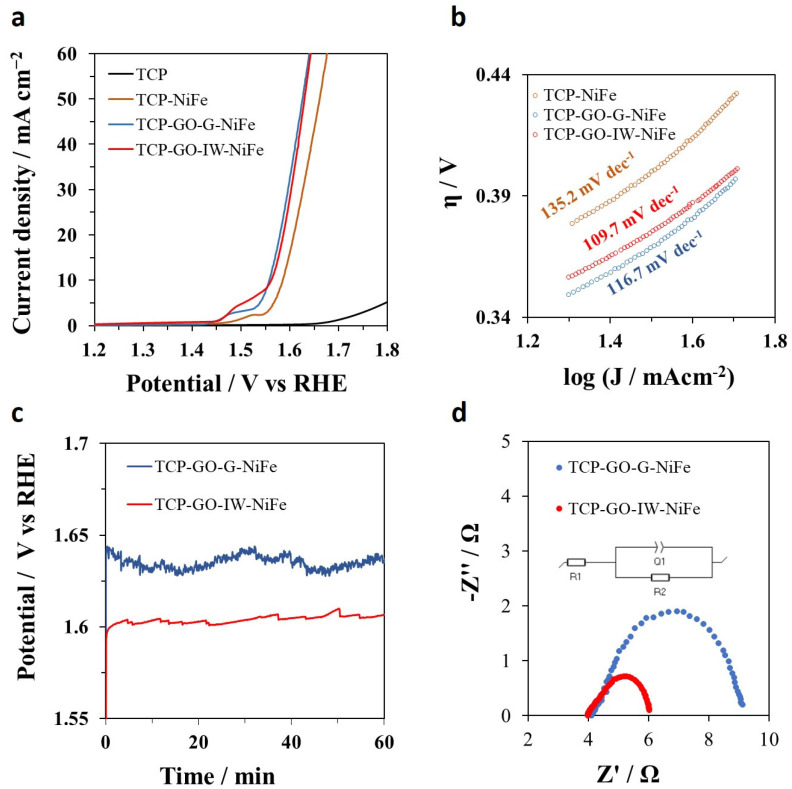
(**a**) Linear sweep voltammetry experiments recorded on the synthetized samples with NiFe and the bare electrode. (**b**) The experiments were carried out at 10 mVs^−1^ in the potential range of 1.2–1.8 V vs. RHE, and the corresponding Tafel slope values were calculated. (**c**) CPs performed at 10 mAcm^−2^ for 1 h. (**d**) Nyquist plots collected in the frequency range between 100 kHz and 100 mHz at 1.6V vs. RHE. All the experiments were carried out in N_2_-saturated KOH 1M electrolyte.

**Table 1 molecules-29-01391-t001:** Main characterization of graphene oxides from waste (GO-IW) and commercial graphite (GO-G).

Sample	XRD			Elemental Analysis	Raman	XPS	
	d_002_ (nm) ^a^	Lc (nm) ^b^	n ^c^	C (Wt.%)	Id/Ig	Csp^2^ (%)	Csp^3^ (%)
GO-G	0.898	9	10	47.5	0.95	43.5	5.6
GO-IW	0.796	11.6	14	49.64	1.07	33.5	7.8

^a^ Interlaminar distance, ^b^ crystalline size, and ^c^ number of layers estimated by XRD as (Lc/d_002_) +1.

**Table 2 molecules-29-01391-t002:** ICP-Ms relative percentage of Ni and Fe in TCP-GO-r-NiFe (r: G and IW).

Sample	Ni (wt.%)	Fe (wt. %)	Ratio Ni/Fe
TCP-GO-G-NiFe	84.3	15.7	5.3
TCP-GO-IW-NiFe	87.2	12.8	6.8

## Data Availability

Data are contained within the article and Appendix A.

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
