# Peer review of "Graphene Materials from Coke-like Wastes as Proactive Support of Nickel–Iron Electro-Catalysts for Water Splitting"

_molecules, 2024, doi:10.3390/molecules29061391_

Round 1

Reviewer 1 Report

Comments and Suggestions for Authors

In this paper, the author study the utilization of the graphene oxide  from this industrial waste in the preparation of 3D electrodes doped with nickel-iron LDH  and their utilization in OER electrocatalytic reactions

1. The table 1 shoule be three-line table.

2. The author should supply the full XPS spectra of TCP-GO-G-NiFe  and  TCP-GO-IW-NiFe  in the Figure 3.

3. The the corresponding Tafel slope values in the Figure 4b are big compared with other report.

4. CPs performed at 10 mAcm-2 for 1 hour.  This time is a bit short. The author shuold supply the 24 hours result.

Author Response

In this paper, the author study the utilization of the graphene oxide  from this industrial waste in the preparation of 3D electrodes doped with nickel-iron LDH  and their utilization in OER electrocatalytic reactions

1. The table 1 should be three-line table.

Table 1 is now a three-line table

2. The author should supply the full XPS spectra of TCP-GO-G-NiFe and TCP-GO-IW-NiFe in the Figure 3.

The Full XPS spectra required are now included in the supplementary materials (Figure S4)

3. The corresponding Tafel slope values in the Figure 4b are big compared with other report.

The Tafel slopes could be a bit larger than other reported, but also with the graphite used as standard, possibly suggesting influence of the support/device. In any case, the results point out the objective of this which is to demonstrate the capability of the graphene materials from a waste to be used as graphene precursor with properties similar to that obtained for a standard graphene oxide.

  1. CPs performed at 10 mAcm-2 for 1 hour.  This time is a bit short. The author should supply the 24 hours result.

As suggested, the CPs at 24h are now included in the Supplementary Materials (Figure S6)

Reviewer 2 Report

Comments and Suggestions for Authors

In this manuscript, González-Ingelmo et al. reported the use of graphene materials from coke-like wastes as proactive support of nickel-iron electro-catalysts for water splitting (oxygen evolution reaction, OER). The electrode made of graphene from wastes showed comparable performance to that made of graphene from graphite, highlighting the promise of using graphene from wastes as efficient electrode materials in electrochemical energy storage and conversion. This can have strong implications in energy sustainability and recycling economy. Overall, this work has good novelty and the manuscript was generally well organized. I feel that this manuscript is suitable for the journal Molecules. However, the below detailed comments need to be addressed to further improve the quality of the manuscript.

1. The authors wanted to showcase that the graphene oxide made of wastes is as effective as that made of graphite in serving as 3D electrode supports for water splitting. To prove this, in their catalyst design, they electrodeposited NiFe LDH onto the support and compared the OER of the materials as-obtained. What about the OER performance of the 3D electrode support without adding NiFe LDH? Did they also show comparable performance?

2. Line 36, recent works on water splitting (Small Methods, 2022, 6, 2201099; InfoMat. 2024, 6, e12494) are recommended to be referenced here.

3. Please address the minor inconsistency in the electrochemical data. In Fig. 4a, both samples TCP-GO-IW-NiFe and TCP-GO-G-NiFe would require a potential of about 1.55-1.56V to reach 10 mA cm-2 (and difference between two samples is minor). However, in Fig. 4c, for the CP tests, they would require much larger potentials (1.6V and 1.63-1.64V), and difference between two samples is a bit larger. Did the authors perform iR correction for these data? This could be one reason for the data inconsistency.

4. Line 42, NiFe based material are promising OER electrocatalysts. Related works can be included (e.g., Nano Energy, 2021, 89, 106463).

5. The EDX spectrum in Fig S1 suggests that the data appears to show high noise. This might be related to the small region of detection. Is there a way to improve?

6. The authors performed the Raman characterisation. However, related figures were not shown, although the ID/IG ratio was given in Table 1.

7. Line 253, there is an error code in the reference area. Please correct.

Author Response

In this manuscript, González-Ingelmo et al. reported the use of graphene materials from coke-like wastes as proactive support of nickel-iron electro-catalysts for water splitting (oxygen evolution reaction, OER). The electrode made of graphene from wastes showed comparable performance to that made of graphene from graphite, highlighting the promise of using graphene from wastes as efficient electrode materials in electrochemical energy storage and conversion. This can have strong implications in energy sustainability and recycling economy. Overall, this work has good novelty and the manuscript was generally well organized. I feel that this manuscript is suitable for the journal Molecules. However, the below detailed comments need to be addressed to further improve the quality of the manuscript.

  1. The authors wanted to showcase that the graphene oxide made of wastes is as effective as that made of graphite in serving as 3D electrode supports for water splitting. To prove this, in their catalyst design, they electrodeposited NiFe LDH onto the support and compared the OER of the materials as-obtained. What about the OER performance of the 3D electrode support without adding NiFe LDH? Did they also show comparable performance?

We have performed the analysis of the 3D electrode support without adding NiFe LDH. The hybrid materials show no substantial catalytic activity (nearly blank experiments). The results are now included in the supplementary material and reference is made in the main text which now reads: “It is also interesting to mention that the 3D systems without electrodeposited Ni and Fe do not show substantial catalytic activity, indicating the poor activity of the graphene materials for themselves (see supplementary material, Figure S5).”

  1. Line 36, recent works on water splitting (Small Methods, 2022, 6, 2201099; InfoMat. 2024, 6, e12494) are recommended to be referenced here.

The new references are now included

  1. Please address the minor inconsistency in the electrochemical data. In Fig. 4a, both samples TCP-GO-IW-NiFe and TCP-GO-G-NiFe would require a potential of about 1.55-1.56V to reach 10 mA cm-2 (and difference between two samples is minor). However, in Fig. 4c, for the CP tests, they would require much larger potentials (1.6V and 1.63-1.64V), and difference between two samples is a bit larger. Did the authors perform iR correction for these data? This could be one reason for the data inconsistency.

A possible explanation of this minor inconsistencies lies in surface phenomena affecting reactivity, such as charge transfer resistance, which is also measured to be different in both materials (Figure 4d). This is now mentioned in the text that now reads:

“Although both samples exhibit stable profiles in CP experiments, there are differences in the required potential to achieve 10mAcm-2 between the two samples, a fact that was not detected in the LSV experiments. These slight variations in favor of TCP-GO-IW-NiFe could be related to differences in surface phenomena affecting reactivity, such as charge transfer resistance. In this sense,”

  1. Line 42, NiFe based material are promising OER electrocatalysts. Related works can be included (e.g., Nano Energy, 2021, 89, 106463).

The works are now included as new references

  1. The EDX spectrum in Fig S1 suggests that the data appears to show high noise. This might be related to the small region of detection. Is there a way to improve?

As stated in the text,it is difficult to improve the SEM EDX measurement. However, we have performed TEM EDX analysis which corroborates the obtained results and are now included in the Supplementary Material (Figure S1,b)

  1. The authors performed the Raman characterisation. However, related figures were not shown, although the ID/IG ratio was given in Table 1.

 The Raman figures are now added in the supplementary material, Figure S7

  1. Line 253, there is an error code in the reference area. Please correct.

The reference mentioned should be 38 now after adding the new references, and is now corrected

Round 2

Reviewer 1 Report

Comments and Suggestions for Authors

The author has revised manuscript according with the review comments.